# Anti-Cancer Effects of Dietary Polyphenols via ROS-Mediated Pathway with Their Modulation of MicroRNAs

**DOI:** 10.3390/molecules27123816

**Published:** 2022-06-14

**Authors:** Yasukiyo Yoshioka, Tomokazu Ohishi, Yoriyuki Nakamura, Ryuuta Fukutomi, Noriyuki Miyoshi

**Affiliations:** 1Graduate School of Integrated Pharmaceutical and Nutritional Sciences, University of Shizuoka, Shizuoka 422-8526, Japan; yoshiokay@u-shizuoka-ken.ac.jp (Y.Y.); yori.naka222@u-shizuoka-ken.ac.jp (Y.N.); 2Institute of Microbial Chemistry (BIKAKEN), Numazu, Microbial Chemistry Research Foundation, Numazu-shi 410-0301, Japan; 3Institute of Microbial Chemistry (BIKAKEN), Laboratory of Oncology, Microbial Chemistry Research Foundation, Shinagawa-ku, Tokyo 141-0021, Japan; 4Quality Management Div. Higuchi Inc., Minato-ku, Tokyo 108-0075, Japan; fukutomi@higuchi-inc.co.jp

**Keywords:** dietary polyphenols, microRNA, cancer, reactive oxygen species, anticancer pathway

## Abstract

Consumption of coffee, tea, wine, curry, and soybeans has been linked to a lower risk of cancer in epidemiological studies. Several cell-based and animal studies have shown that dietary polyphenols like chlorogenic acid, curcumin, epigallocatechin-3-*O*-gallate, genistein, quercetin and resveratrol play a major role in these anticancer effects. Several mechanisms have been proposed to explain the anticancer effects of polyphenols. Depending on the cellular microenvironment, these polyphenols can exert double-faced actions as either an antioxidant or a prooxidant, and one of the representative anticancer mechanisms is a reactive oxygen species (ROS)-mediated mechanism. These polyphenols can also influence microRNA (miR) expression. In general, they can modulate the expression/activity of the constituent molecules in ROS-mediated anticancer pathways by increasing the expression of tumor-suppressive miRs and decreasing the expression of oncogenic miRs. Thus, miR modulation may enhance the anticancer effects of polyphenols through the ROS-mediated pathways in an additive or synergistic manner. More precise human clinical studies on the effects of dietary polyphenols on miR expression will provide convincing evidence of the preventive roles of dietary polyphenols in cancer and other diseases.

## 1. Introduction

Human epidemiological studies have shown that diets high in plant polyphenols have beneficial effects on various diseases including cancer [1,2]. We have discussed the anticancer effects of coffee, tea, wine, and curry based on recent evidence from human studies, in which chlorogenic acid (CGA), (-)-epigallocatechin gallate (EGCG), resveratrol (RES), and curcumin (CUR), respectively, are believed to be major contributors to the activity [3] (Figure 1 and Table 1).

Quercetin (QUE) is a flavonol found in a variety of fruits and vegetables including apples, grapes, broccoli, green tea, and onions [4,5] (Figure 1), and several human studies have shown that QUE-rich diets have anticancer effects [5,6,7,8]. For example, Ekström et al. [7] discovered that QUE intake had a strong inverse association with the risk of noncardia gastric adenocarcinoma, with an adjusted odds ratio (OR) of 0.57 (95% confidence interval [CI] = 0.40–0.83) when the highest quintile (≥11.9 mg/day) was compared to the lowest quintile (<4 mg).

Epidemiologic studies have also shown that a soy-rich diet reduces the risk of various diseases, including cancer, and one of the main contributors is thought to be genistein (GEN), a phenolic compound [9,10,11] (Figure 1). Wang et al. [12] discovered a lower risk of papillary macrocarcinomas in women who consumed 1860–3110 μg/day of GEN (OR = 0.26, CI = 0.08–0.85) compared to women who consumed <760 μg/day in a population-based case-control study in Connecticut from 2010 to 2011. A meta-analysis conducted by Applegate et al. [13] revealed that the pooled relative risk for GEN in the risk of prostate cancer was 0.90 (CI: 0.84–0.97).

Many epidemiological studies, on the other hand, have found that these foods have no anticancer effects [1,14]. The inconsistent results could be due to a number of confounding factors, including the quantity and quality of plant foods consumed, as well as residual pesticides and acrylamide formed during preparation, cigarette smoking, alcohol consumption, differences in ingredients, hormonal activities, microbiota, and genetic background [1,14,15]. Human intervention studies that are well-designed could provide significant evidence for the anticancer effects of dietary foods containing these polyphenols.

The anticancer properties of these polyphenols have been demonstrated in a large number of cell-based and animal studies, and their possible anticancer mechanisms have been proposed. Of them, one involving reactive oxygen species (ROS) appears to be the most likely, in which these polyphenols can act as both an ROS-generator and an ROS-scavenger [16].

In our previous review, we presented putative anticancer pathways that CGA, CUR, EGCG, or RES can trigger [3], as well as the roles of microRNAs (miRs) modulated by these polyphenols in the pathways. As GEN and QUE share some properties similar to CGA, CUR, EGCG, and RES, this review focuses on their ROS-mediated anticancer properties, which may include their miR-modulating activity.

## 2. Anticancer Pathways

Based on our previous discussions [3,17,18], Figure 2 depicts a putative ROS-mediated anticancer mechanism in which polyphenols may be involved [19,20,21,22,23,24,25,26,27,28,29,30,31,32]. Based on the findings of Zhang et al. [33], a pathway involving AMP-activated protein kinase (AMPK), SIRT1, p53, and p21 is depicted in this figure. They discovered that *S*-nitrosoglutathione, an endogenous nitric oxide carrier, induces apoptosis in lung cancer A549 cells by inhibiting SIRT1 deacetylase activity toward p53 and thus increasing p53 acetylation, which leads to an increased expression of p21 and apoptosis in A549 cells. According to our previous discussion [3], links of miRs to the constituting molecules in the pathways are also presented in Figure 2.

As shown in Table 2 [14,21,34,35,36,37,38,39,40,41,42,43,44,45,46,47,48,49,50,51,52,53,54,55,56,57,58,59,60,61,62,63,64,65,66,67,68,69,70,71,72,73,74,75,76,77,78,79,80,81,82,83,84,85,86,87,88,89,90,91,92,93,94,95,96], these six polyphenols are similar in that they can act as an ROS-generator and an ROS-scavenger, respectively, leading to AMPK upregulation and NF-κB downregulation. GEN and QUE also influence other molecular components of the anticancer pathways depicted in Figure 2 and Table 3 [50,97,98,99,100,101,102,103,104,105,106,107,108]. At present, it is not clear what can direct a polyphenol to act as an ROS-generator or ROS-scavenger. Differences in cell types, concentrations of polyphenols and metal ions such as Fe(II) and Cu(II); antioxidant enzymes such as glutathione *S*-transferase, glutathione peroxidase, and hemeoxygenase-1; and small molecules such as glutathione [18,109] are all possible candidates.

miRs in red and in blue are upregulated and downregulated by polyphenols, respectively.

For example, Kanadzu et al. [110] demonstrated a concentration-dependent dual function of EGCG by showing that EGCG at 1–100 µM enhanced DNA strand breakage induced by bleomycin and hydrogen peroxide, whereas a lower concentration at 0.1 to 0.01 µM suppressed DNA breakage in human lymphocytes. CUR was shown to increase superoxide production in MCF-7, HepG2, and MDAMB cancer cells, but not in normal rat hepatocytes [111]. Low concentrations of GEN promoted primary muscle cell proliferation, whereas high concentrations inhibited their proliferation by causing intracellular ROS production [112].

## 3. Modulation of miRs by Dietary Polyphenols

Polyphenols can influence the expression of miRs, which are 20–22 nucleotide long single-stranded non-coding RNAs [3]. As miRs regulate a wide range of biological processes, including cell proliferation, apoptosis, and cell differentiation, changes in their expression levels are linked to disease progression, including cancer [113]. When compared to normal cells or tissues, the expression of miRs is upregulated (oncogenic miRs) or downregulated (tumor suppressor miRs) in cancers, indicating their important roles in cancer.

We previously discussed the modulatory activity of CGA, CUR, EGCG, and RES [3], and dietary polyphenols can affect miR expression. At least three of these polyphenols can modulate the same nine miRs, five of which are downregulated (miR-20a, 21, 25, 93, and 106b) and four of which are upregulated (miR-16, 34a, 145, and 200c). Based on our previous discussion [3] and information on caspase 3 [114], we depict Figure 2 for the ROS-mediated anticancer pathways. As mentioned earlier, GEN and QUE share many similar properties with the other four polyphenols, implying that they have similar miR-modulatory effects. Table 4 and Table 5 compare the effects of these dietary polyphenols on miRs reported in the literature [21,23,24,26,27,29,30,31,32,115,116,117,118,119,120,121,122,123,124,125,126,127,128,129,130,131,132,133,134,135,136,137,138,139,140,141,142,143,144,145,146,147,148,149,150,151,152,153,154,155,156,157,158,159,160,161,162,163,164,165,166,167,168,169,170,171,172,173,174,175,176,177,178,179,180,181,182]. The addition of GEN and QUE data increased the number of miRs modulated similarly by at least three polyphenols from 9 to 15, as expected. The effects of miRs modulated by these polyphenols on the molecular constituents in the ROS-mediated pathways are also provided in these tables and incorporated in Figure 2.

## 4. Anticancer Mechanism of Tumor Suppressor miRs Upregulated by Polyphenols

Table 4 summarizes the available data for tumor-suppressor miRs that are commonly upregulated by at least three different polyphenols in cancer cells. Figure 2 shows that several molecules involved in the anticancer mechanism are found in ROS-mediated pathways. Table 4 also provides information on the modulatory effects of miRs upregulated by these polyphenols on these molecules.

**Table 4 molecules-27-03816-t004:** Tumor-suppressor miRs upregulated by polyphenols, cell types examined, and effects of miR upregulation.

miR	CUR	EGCG	GEN	QUE	RES	Effects of miRs Upregulated by Polyphenols on Molecules in the ROS-Mediated Pathway: ↑, Upregulation; ↓ Downregulation
**miR-16**	MCF-7 (breast cancer) (Yang, et al.) [182]	HepG2 (liver cancer) (Tsang, et al.) [115]		A549 (lung cancer) (Sonoki, et al.) [116] HSC-6 SCC-9 (oral cancer) (Zhao, et al.) [117]	MCF7-ADR MCF10A MDA-MB-231-luc-D3H2LN (breast cancer) (Hagiwara, et al.) [118] CCRF-CEM (acute lymphoblastic leukemia) (Azimi, et al.) [119]	↓Bcl-2 [115,182]
**miR-22**	BxPC-3 (pancreatic carcinoma) (Sun, et al.) [120] Y79 (retinoblastoma) (Sreenivasan, et al.) [121] *Downregulated ** *MyLa2059,* *SeAx* (*malignant cutaneous lymphoma*) (Sibbesen, et al.) [122]	CNE2 (nasopharyngeal carcinoma) (Li, et al.) [123]		Tca8113 SAS (oral squamous cell carcinoma) (Zhang, et al.) [124]		↓VEGF via↓Sp1 [120]
**miR-34a**	MDA-MB-231 MDA-MB-435 (breast cancer) (Guo, et al.) [125] SGC-7901 (gastric cancer) (Sun, et al.) [126] HCT116 (colorectal cancer) (Toden, et al.) [127] BxPC-3 (pancreatic cancer) (Sun, et al.) [120] *Downregulated ** *TE-7* (*esophageal adenocarcinoma*) (Subramaniam, et al.) [128]	SK-N-BE2 IMR-32 (malignant neuroblastoma) (Chakrabarti, et al.) [129] SH-SY5Y SK-N-DZ (malignant neuroblastoma) (Chakrabarti, et al.) [130] HCT116 HCT116-5FUR (colorectal cancer, 5FU resistant) (Toden, et al.) [131] CNE2 (nasopharyngeal carcinoma) (Li, et al.) [123] HepG2 (hepatocellular carcinoma) (Mostafa, et al.) [132]	HNC-TICs (tumor-initiating cells of head and neck cancer) (Hsieh, et al.) [133] DU145 (prostate cancer) (Chiyomaru, et al.) [134] AsPC-1 MiaPaCa-2 (pancreatic cancer) (Xia, et al.) [135]		MDA-MB-231-luc-D3H2LN (breast cancer) (Hagiwara, et al.) [118] DLD-1 (colon cancer) (Kumazaki, et al.) [136] MCF-7 (breast cancer) (Otsuka, et al.) [137] SKOV-3 OV-90 (ovarian cancer) (Yao, et al.) [138]	↓Bcl-2 [125,126,127,138] ↓NF-κB via Notch-1 [135]
**miR-141**	HCT116-5FUR (colorectal cancer, 5FU resistant) (Toden, et al.) [139]	*Downregulated ** *MM1.s* (*multiple myeloma*) (Gordon, et al.) [140]	786-O ACHN (renal carcinoma) (Chiyomaru, et al.) [141]		MCF7-ADR MCF-7 MCF10A MDA-MB-231-luc-D3H2LN (breast cancer) (Hagiwara, et al.) [118]	
**miR-145**	U-87 MG (glioblastoma) Mirgani, et al.) [142] DU145 22RV1 (prostate cancer) (Liu, et al.) [143]	HCT116 HCT116-5FUR (colorectal cancer, 5FU resistant) (Toden, et al.) [131]	Y79 (retinoblastoma) (Wei, et al.) [144]	SKOV-3 A2780 (ovarian cancer) (Zhou, et al.) [145]	BT-549 MDA-MB-231 MCF-7 (breast cancer) (Sachdeva, et al.) [146]	↑Caspase-3 [145]
**miR-146a**	U-87 MG (glioblastoma) (Wu, et al.) [31] AsPC-1 (pancreatic cancer) CDF (analog) (Bao, et al.) [147]		Colo357 Panc-1 (pancreatic cancer) G2535 (mixture of genistein and other isoflavones) (Li, et al.) [148]	MCF-7 MDA-MB-231 (breast cancer) (Tao, et al.) [26]		↓NF-κB [31] ↑Caspase-3 [26] ↓EGFR [26]
**miR-200c**	HCT116-5FUR SW480-5FUR (colorectal cancer, 5FU resistant) (Toden, et al.) [139] MiaPaCa-2 MiaPaCa-2-GR BxPC-3 (pancreatic cancer) CDF (analog) (Soubani, et al.) [149]	HCT116-5FUR (colorectal cancer, 5FU resistant) (Toden, et al.) [131]			Cancer stem cells of nasopharyngeal carcinoma (Shen, et al.) [150] MCF7-ADR MCF-7 MCF10A MDA-MB-231-luc-D3H2LN (breast cancer) (Hagiwara, et al.) [118] HCT116 (colorectal cancer) (Dermani, et al.) [151]	↑PTEN [149]

* The items shown in italics are different findings from other reported results (see Text).

### 4.1. miR-16

CUR, EGCG, QUE, and RES have been shown to have anticancer properties [3,17,183]. These polyphenols have been shown to increase the expression of the tumor suppressor miR-16. miR-16 has the ability to reduce the expression of the target Bcl-2 [115]. Claudin-2 expression is decreased by QUE-induced miR-16, which may downregulate Bcl-2 [116]. Bcl-2 is an anti-apoptotic protein, and its inhibition would result in an anticancer effect. QUE may increase miR-16 expression to decrease Homeobox A10 expression, which is involved in cancer proliferation, migration, and invasion [117]. RES increased the expression and activity of Argonaute2, a central RNA interference component, which resulted in anticancer effects by increasing the expression of several tumor-suppressor miRs including miR-16 [118].

### 4.2. miR-22

CUR, EGCG, and QUE have been shown to upregulate miR-22, which may downregulate specificity protein 1 (Sp1)**,** estrogen receptor 1 (ESR1) [120], erythoblastic leukemia viral oncogene homolog 3 (Erbb3) [121], and nuclear receptor coactivator 1 (NCoA1) [122]. Sun et al. [120] discovered that CUR increased miR-22 expression in PxBC-3 pancreatic cancer cells using oligonucleotide microarray analysis. Transfection with miR-22 mimetics reduced expression of the target genes Sp1 and ESR1, whereas antisense inhibition of miR-22 increased Sp1 and ESR1 expression. Sp1 is overexpressed in various cancers and has the potential to be a chemotherapeutic drug target [184]. Sp1 can upregulate VEGF to promote cancer cell growth, angiogenesis, and metastasis [185,186], downregulation of miR-22 upregulated by these polyphenols may contribute to the anticancer effects of these polyphenols.

In malignant T cells, transfection of recombinant miR-22 resulted in the inhibition of its targets including NCoA1, HDAC6, MAX, MYCBP, and PTEN [122]. As PTEN is known to be tumor suppressing [187], its downregulation by CUR does not appear to be consistent with CUR’s anticancer properties. Downregulation of other cancer-promoting molecules such as HDAC6, required for efficient oncogenic tumorigenesis [188], and NCoA1, whose overexpression increases the number of circulating cancer cells and the metastasis [189], may overwhelm PTEN’s efficacy in this case.

Zhang et al. [124] showed that overexpression of miR-22 increased cancer cell apoptosis by targeting WNT1, and that the miR-22/WNT1/β-catenin axis is the downstream pathway for QUE to exert an antitumor effect in oral squamous cell carcinoma.

### 4.3. miR-34a

CUR upregulation of miR-34 resulted in Bcl-2 downregulation, cell cycle arrest, and/or c-Myc downregulation [125,126,127]. RES increased apoptosis and miR-34a expression in ovarian cancer cells [138]. miR-34a inhibition experiments revealed that miR-34a downregulates Bcl-2, upregulates Bax, and activates caspase-3.

EGCG has been shown to exert anticancer effects by upregulating tumor-suppressing miRs including miR-34a and downregulating oncogenic miRs such as miR-92, miR-93, and miR-106b [130].

In an experiment with HNC-TICs cells from head and neck cancer, GEN inhibited their proliferation, downregulated epithelial–mesenchymal transition (EMT), and induced upregulation of miR-34a, which resulted in ROS production [133]. Caspase-3 activation induced by overexpression of miR-34a was inhibited by *N*-acetylcysteine, indicating that ROS are involved in the anticancer effects of GEN.

In, GEN induced apoptosis in prostate cancer PC3 and DU145 cells, increased miR-34a expression levels, and reduced those of oncogenic HOX transcript antisense RNA (HOTAIR), a target of miR-34a [134]. HOTAIR is a non-coding RNA that has been shown to induce cell cycle arrest in the G_2_/M phase [190]. The GEN-mediated upregulation of miR-34a in pancreatic cancer cells also inhibited the Notch-1 signaling pathway [135], whose activation promotes cancer cell growth and metastasis [191,192]. Inhibition of Notch-1 would result in down regulation of NF-κB, leading to cancer suppression [193].

RES increased the expression of tumor suppressor miR-34a, 424, and 503 in breast cancer cells [137]. HNRNPA1, a heterogeneous nuclear ribonucleoprotein associated with tumorigenesis and progression, was directly downregulated by miR-424 and miR-503, but indirectly by miR-34a [137]. According to Kumazaki et al. [136], RES upregulates miR-34a, which causes downregulation of the target gene E2F3 and its downstream SIRT1, leading to inhibition of colon cancer cell growth.

Thus, polyphenols appear to upregulate miR-34 in general, but Subrama-niam et al. [128] found that CUR decreased expression of miR-34a in esophageal cancer TE-7 cells. One possible explanation for the difference is that the p53 status of different cell lines differs, as TE-7 cells are p53-deficient and p53 is an upstream regulator of miR-34a.

### 4.4. miR-141

CUR upregulated the expression of EMT-suppressing miRs such as miR-34a, 101, 141, 200c, and 429 in 5-fluorouracil (5FU)-resistant HCT116 cells, but not in 5FU-resistant SW480 cells [139]. EMT is a crucial step in the generation of cancer stem cells and the progression of cancer. The extent to which miR-141 contributes to EMT suppression is not known.

Chiyomaru et al. [141] discovered that treatment of renal carcinoma cells with GEN increased miR-141 expression and decreased HOTAIR, which is known to promote malignancy. HOTAIR expression was reduced in cells transfected with pre-miR-141. By increasing the expression of a number of tumor-suppressive miRs, including miR-16, 141, 143, and 200c, RES reduced the viability of breast cancer cells and inhibited cancer stem-like cell characteristics [118]. The miR-141 inhibitor reduced the efficacy of RES’s inhibitory effect against cancer invasion, implying that miR-141 plays a role in RES’ anticancer effect.

Gordon et al. [140] reported that treatment of multiple myeloma, MM1.s cells, with the carcinogen benzo[a]pyrene upregulated the expression of miR-15a, 16, 25, 92, 125b, 141, and 200a, all of which are p53 targets. EGCG inhibited the expression of tumor-suppressive miR-141 which upregulates p53. The finding appears inconsistent with EGCG’s anticancer activity. It is possible that EGCG’s downregulation of oncogenic miR-25 may be more effective in the anticancer effect than downregulation of miR-141 in these cells.

### 4.5. miR-145

Curcumin encapsulated in a non-toxic nanocarrier inhibited the proliferation of glioblastoma U-87 MG cells, increased miR-145 expression, and decreased the expression of transcription factors Oct4, SOX-2, and Nanog, all of which are upregulated and result in increased metastasis, invasion, and recurrence [142,194].

CUR inhibited the proliferation, invasion, and tumorigenicity of prostate cancer stem cells HuPCaSCs (CD44^+^/CD133^+^ subpopulation isolated from prostate cancer cell lines Du145 and 22RV1) by increasing the expression of miR-145, which prevents cell proliferation by decreasing Oct4 expression [143]. In colorectal cancer cells, EGCG increased apoptosis and cell cycle arrest, and upregulated miR-145 [131].

In GEN-treated retinoblastoma Y79 cells, miR-145 was found to be significantly upregulated [144]. The siRNA downregulated miR-145 and the target of miR-145 has been identified as ABCE1 which has oncogene-like properties. By increasing the expression of miR-145, QUE was found to induce apoptosis in human ovarian carcinoma cells. The increased expression levels of cleaved caspase-3 induced by QUE were further increased by overexpression of miR-145 [145].

### 4.6. miR-146a

CUR upregulated miR-146a in human U-87 MG glioblastoma cells, and overexpression of miR-146a increased apoptosis and decreased NF-κB activation in cells treated with the anticancer drug temozolomide [31]. miR-146a expression is lower in pancreatic cancer cells compared to normal human pancreatic duct epithelial cells. GEN treatment increased miR-146a expression with decreasing EGFR and NF-κB expression in these cancer cells. Transfection of miR-146a inhibited these cells’ invasive ability by downregulating EGFR and NF-κB, implying that upregulation of miR-146a is involved in the anticancer effect of GEN [148]. The results of experiments with or without transfection of miR-146a mimic or anti-miR-146a revealed that QUE increased miR-146a, leading to apoptosis induction through downregulation of EGFR and activation of caspase-3 in a study of QUE’s anticancer effect [26].

### 4.7. miR-200c

Experiments on overexpression or silencing of miR-200c in pancreatic cancer MiaPaCa-2 cells showed that a CUR analog upregulated PTEN expression, increased levels of MT1-MMP, and reduced tumor cell aggressiveness through upregulation of miR-200c [149]. Toden et al. [139] discovered that CUR improved the efficacy of 5-FU in suppressing tumor growth and EMT in 5FU-resistant colorectal cancer cells. miR-200c, a key EMT-suppressing miR, was upregulated by CUR, and miR-200c was found to downregulate BMI1, SUZ12, and EZH2 in a transfection experiment.

Upregulation of miR-200c was also observed in RES-treated nasopharyngeal carcinoma cancer stem cells [150], EGCG-treated 5FU-resistant colorectal cancer cells [131], and RES-treated breast cancer cells [118]. Dermani et al. [151] discovered that RES increased the expression of miR-200c and decreased the viability of colorectal cancer cells. Transfection with anti-miR-200c increased vimentin and ZEB1 expression, while decreasing E-cadherin expression and apoptosis. These changes were reversed by RES, indicating that RES induces apoptosis and inhibits EMT in colorectal cancer by regulating miR-200c.

## 5. Anticancer Mechanism of Oncogenic miRs Downregulated by Polyphenols

Table 5 summarizes the available data for oncogenic miRs that are commonly modulated by at least three different polyphenols in cancer cells. Among these molecules, Figure 2 shows that several molecules involved in the anticancer mechanism are found in the ROS-mediated pathways. Table 5 also shows the effects of miRs downregulated by polyphenols on the molecules involved in ROS-mediated anticancer pathways (Figure 2).

### 5.1. miR-20a

CGA inhibited hepatoma and lung cancer cells by causing cell cycle arrest in the G_0_/G_1_ phase [152]. CGA increased KHSRP, p53, and p21 expression while decreasing c-Myc and CD44 expression. The microarray analysis revealed that the expression of the miR-17 family members miR-20a, 93, and 106b was downregulated in cells treated with CGA. An inhibitor of miR-20a increased p21 mRNA expression, and transfection of CGA-treated cells with a mimic of miR-20a which cancelled CGA’s p21 upregulation effect while increasing c-Myc, indicating that p21 is the miR’s target.

Dhar et al. [156] discovered that RES reduced the expression of miRs-17, 20a, 106a, and 106b in prostate cancer cells. In an extended study, they discovered that RES downregulation of these miRs increased the expression of their target PTEN. These miRs, when expressed ectopically, directly targeted PTEN 3’UTR, leading to the reduction of its expression [154].

Liver fibrosis is often linked to the development of cancer [157]. RES was shown to attenuate liver fibrosis in an animal model in a study to investigate its role in this pathology. Cell-based experiments with an miR-20a mimic revealed that RES induces autophagy and activates the miR-20a-mediated PTEN/PI3K/AKT signaling pathway, resulting in fibrosis prevention [155].

### 5.2. miR-21

Increases in the mRNA levels of miR-21 and connective tissue growth factor (CTGF) and a decrease in the level of Smad7 were caused by IL-13 stimulation of LX-2 cells, which were reversed by CGA [195]. miR-21 knockdown resulted in lower mRNA levels of miR-21 and CTGF expression, while Smad7 levels increased in line with the findings on the protein expression levels of CTGF, p-Smad1, p-Smad2, p-Smad2/3, and TGF-β receptor 1. The affected tissues had increased mRNA levels of miR-21 and CTGF with a decrease in the level of Smad7 and CGA, which prevented these changes and liver fibrosis in an animal model of liver fibrosis induced by *Schistosoma japonicum cercaria* infection. Since liver fibrosis is intimately related to liver cancer, these findings suggest anticancer effects of CGA as well [157].

**Table 5 molecules-27-03816-t005:** miRs downregulated by polyphenols, cell types examined, and effects of miR downregulation.

miR	CGA	CUR	EGCG	GEN	QUE	RES	Effects of miRs Downregulated by Polyphenols on Molecules in the ROS-Mediated Pathway: ↑, Upregulation; ↓, Downregulation
**miR-20a**	Huh7 (Hepatoma) H446 (lung carcinoma) (Huang, et al.) [152]	RKO (colon cancer) (Gandhy, et al.) [27]	HUVEC (umbilical vascular endothelial cell cocultured with A549) (Mirzaaghaei, et al.) [153]			DU145 22RV1 (prostate cancer) (Dhar, et al.) [154] (CCL_4_-induced liver fibrotic cells) (Zhu, et al.) [155] DU145 (prostate cancer) (Dhar, et al.) [156]	↑ p21 [152] ↑ PTEN [154] ↑ PTEN/PI3K/AKT [155]
**miR-21**	LX2 (hepatic stellate) (Wang, et al.) [157]	HCT116 RKO (colorectal cancer) (Mudduluru, et al.) [158] AsPC-1 MiaPaCa-2 (pancreatic cancer) CDF (analog) (Bao, et al.) [147] TE-7 (esophageal cancer) (Subramaniam, et al.) [128] PC-3 LNCaP (prostate cancer) Hypoxia CDF (analog) (Bao, et al.) [23] A549 (lung cancer) (Zhang, et al.) [159] K562 LAMA84 (chronic myelogenous leukemia) (Taverna, et al.) [160] DU145 C4-2 (prostate cancer) (Yallapu, et al.) [161]	MCF-7 (breast cancer) Polyphenon-60 (Fix, et al.) [162] 22Rv1 xenograft (prostate tumor) (Siddiqui, et al.) [163]	A-498 xenograft (renal cancer) (Zaman, et al.) [164]		SW480 (colon cancer) (Tili, et al.) [165] PC-3M-MM2 (prostate cancer) (Sheth, et al.) [166] PANC-1 CFPAC-1 MiaPaCa-2 (pancreatic cancer) (Liu, et al.) [167] U251 (glioblastoma) (Li, et al.) [32] T24 5637 (bladder cancer) (Zhou, et al.) [29]	↓VEGF [23] ↓IL-6 [23] ↑PTEN [159,160] ↑p21 [164] ↓Bcl-2 [29,167] ↓NF-κB [32] ↓Akt [29]
**miR-25**		BxPC-3 (pancreatic cancer) (Sun, et al.) [120]	MCF-7 (breast cancer) Polyphenon-60 (Fix, et al.) [162] MM1.s (multiple myeloma) (Gordon, et al.) [140] MCF-7 (breast cancer) (Zan, et al.) [168]			SW480 (colon cancer) (Tili, et al.) [165]	↑ p53 [140] ↑Caspase-3 [168]
**miR-27a**		HCT116p53± SW480 (Toden, et al.) [127] SW480 (colon cancer) (Noratto, et al.) [169] RKO (colon cancer) (Gandhy, et al.) [27]	MCF-7 (breast cancer) Polyphenon-60 (Fix, et al.) [162]	PANC-1 BxPC-3 (pancreatic cancer) (Cheng, et al.) [170] SKOV3 (ovarian cancer) (Xu, et al.) [171] C918 (uveal melanoma) (Sun, et al.) [172] *Upregulated* * *A549* (lung cancer) (Yang, et al.) [173]			↓VEGF via Sp1 [169] ↓VEGF via Sp1 [27] ↓EGFR [27] ↓Survivin [27] ↓Bcl-2 [27] ↓NF-κB [27] ↑FOXO1 [170]
**miR-93**	Huh7 (Hepatoma) H446 (Lung carcinoma) (Huang, et al.) [152]		SK-N-BE2 IMR-32 (malignant neuroblastoma) (Chakrabarti, et al.) [129] SH-SY5Y SK-N-DZ (malignant neuroblastoma) (Chakrabarti, et al.) [130]			MCF-10A (breast cancer) (Singh, et al.) [174]	↑ p21 [152] ↑Caspase-3 [129,130]
**miR-106b**	Huh7 (Hepatoma) H446 (Lung carcinoma) (Huang, et al.) [152]		SK-N-BE2 IMR-32 (malignant neuroblastoma) (Chakrabarti, et al.) [129] SH-SY5Y SK-N-DZ (malignant neuroblastoma) (Chakrabarti, et al.) [130]			LNCaP DU145 (prostate cancer) (Dhar, et al.) [156] DU145 22RV1 (prostate cancer) (Dhar, et al.) [154]	↑ p21 [152] ↑ PTEN [154,156]
**miR-155**	RAW264.7 (mouse macrophage) (Zeng, et al.) [21]	RAW264.7 (mouse macrophage) THP1 (acute monocyte leukemia) (Ma, et al.) [30]		MDA-MB-435 Hs578t (breast cancer) (Parra, et al.) [175] (Basu, et al.) [196]	RAW264.7 (mouse macrophage) (Boesch-Saadatmandi, et al.) [176]	THP-1 (monocyte) (Tili, et al.) [177]	↓ NF-κB [21] ↑ PTEN [175]
**miR-221**		MiaPaCa-2 (pancreatic cancer) CDF (analog) (Sarkar, et al.) [178] HepG2 tumor (HCC orthotopic mouse model) (Zhang, et al.) [24] SW1736 (Anaplastic thyroid carcinoma) (Allegri, et al.) [179]	SW1736 (Anaplastic thyroid carcinoma) (Allegri, et al.) [179] *Upregulated* * *HepG2* (liver cancer) (Tsang, et al.) [115]	PC-3 (prostate cancer) (Chen, et al.) [180] SW1736 (Anaplastic thyroid carcinoma) (Allegri, et al.) [179]	WI-38 (lung fibroblast) (Wang, et al.) [181]		↑ PTEN [178] ↓VEGF [24]

* The items shown in italics are different findings from other reported results (see Text).

CUR inhibited colorectal cancer cell proliferation by inducing G_2_/M arrest [158]. CUR inhibited AP-1 binding to the promoter of miR-21 and induced the expression of the tumor suppressor programmed cell death protein 4, which is a target of miR-21.

In pancreatic cancer AsPC-1 and MiaPaCa-2 cells, Bao et al. [147] discovered that a CUR analog CDF suppressed the expression of histone methyltransferase EZH2, EpCAM, ABCG2, Shh, MMP-9, cleaved Notch-1, and Hes-1, while increasing the miR expressions of let-7 family miRs, miR-26a, 101, 146a, and 200. The expression of miR-21 was extremely high in these cells, and CDF suppressed its expression. The same group of researchers also discovered that hypoxia increases the expression of VEGF, IL-6, and CSC marker genes such as Nanog, Oct4, and EZH2, as well as the expression of miR-21 in prostate cancer cells [23].

CUR inhibited esophageal cancer cell proliferation and colony formation by inducing apoptosis through caspase 3 activation [128]. CUR also inhibited Notch-1 activation, Jagged-1 expression and its downstream target Hes-1, as well as downregulation of miR-21 and miR-34a expression and upregulation of tumor suppressor let-7a miR.

CUR inhibited cell proliferation, induced apoptosis and suppressed miR-21 expression in A549 cells. PTEN, a putative miR-21 target, was upregulated by CUR. miR-21 transfection suppressed CUR’s effects on cell proliferation and apoptosis in these cells, suggesting that miR-21 suppression may have anticancer therapeutic benefits [159]. Similarly, CUR reduced cell viability and miR-21 expression in chronic myelogenous leukemia cells [160]. PTEN was upregulated by CUR, while VEGF was downregulated. miR-21 mimic transfection increased VEGF expression, while miR-21 inhibitor decreased VEGF expression. CUR reversed the effect of a miR-21 mimic, while increasing the effect of an miR-21 inhibitor, indicating that VEGF is a target of miR-21 in CUR’s anticancer effects.

CUR inhibited cell growth and miR-21 expression in prostate cancer cells [161]. Western blot analysis showed that CUR caused increased levels of the cleaved PARP, and decreased levels of Bcl-xL, Mcl-1, and p-Akt, respectively.

Polyphenon -60, which contains EGCG as a major component, caused downregulation of miR-21 expression, which can downregulate the tumor suppressor gene tropomyosin-1 in MCF-7 breast cancer cells [162]. EGCG inhibits prostate cancer cell growth. The tumor xenograft tissues from EGCG-treated mice had decreased levels of miR-21 and increased levels of miR-330 [163].

Zaman et al. [164] discovered that GEN inhibited tumor formation by inhibiting miR-21 expression in kidney cancer A-498 cells and xenografts. Inhibition of cell growth, induction of G0/G1 arrest, and upregulation of p21 and p38 MAP kinase were all observed when miR-21 was knocked down in these cells, indicating that p21 could be a target of miR-21.

A microarray analysis showed that RES downregulated several oncogenic miRs including miR-21 and upregulated tumor-suppressing miRs, including miR-663 in colon cancer cells, suggesting that RES’ anticancer effects may be influenced by changes in the composition of miR populations in cancer cells [165]. A similar study in prostate cancer cells revealed that RES reduced the expression of miR-21, which was confirmed by qRT-PCR [166]. Transfection with pre-miR-21 resulted in the downregulation of tumor-suppressing PDCD4 and the upregulation of cancer cell invasion, which were both reversed by RES.

RES reduced the viability of pancreatic cancer cells and suppressed miR-21 expression [167]. Bcl-2 expression was reduced when miR-21 expression decreased. Transfection of a miR-21 mimic reversed RES-induced downregulation of Bcl-2 and apoptosis, indicating that miR-21 is a target of the RES’s anticancer action. Similar results were reported for RES’s anticancer effects in bladder cancer cells [29]. miR-21 overexpression attenuated the inhibition of p-Akt activity and downregulated Bcl-2 expression and apoptosis induced by RES. Furthermore, RES’s anticancer mechanism against glioma cells was reported to involve miR-21 [32]. RES decreased IκB phosphorylation, nuclear p65 protein levels, and NF-κB activity. miR-21 expression was inhibited by RES, and miR-21 downregulation reduced NF-κB activity. The effect of RES on NF-κB activity and apoptosis was reversed when miR-21 was overexpressed.

### 5.3. miR-25

According to a microarray analysis, CUR-treated pancreatic cancer cells had lower expression of miR-25 and other miRs than untreated pancreatic cancer cells [120]. In colon cancer cells, RES was found to downregulate several oncogenic miRs, including miR-25 [165]. Fix et al. [162] showed that breast cancer cells treated with Polyphenon-60 exhibited upregulated expression of let-7a, 107, 548m, 720, 1826, 1978, and 1979 and downregulated expression of let-7c, let-7e, let-7g, miR-21, 25, 26b, 27a, 27b, 92a, 125a-5p, 200b, 203, 342-3p, 454, 1469, and 1977.

Gordon et al. [140] discovered that the carcinogen benzo[a]pyrene upregulated the expression of p53-targeting miRs including miR-25 in MM1.s cells. EGCG inhibited the expression of miR-25 in these cells as well as the induction of miR-25 by the carcinogen, suggesting that miR-25 is involved in EGCG’s anticarcinogenic activity. In breast cancer cells, Zan et al. [168] discovered that EGCG inhibited miR-25 expression, as well as induction of apoptosis and disruption of cell cycle progression at G_2_/M phase. The apoptotic effects of EGCG, such as caspase-3 and caspase-9 activation and an increase in PARP expression, were reduced when cells were transfected with miR-25 mimic.

### 5.4. miR-27a

CUR inhibited the expression of miR-27a and had cytotoxic effects on colorectal cancer cells [127]. In colorectal cancer cells, knockdown of miR-27a increased apoptosis and G2/M phase arrest. Curcuminoids inhibited the growth of colon cancer cells and suppressed miR-27a while downregulating Sp1, Sp3, and Sp4 and Sp-regulated genes [169]. Treatment of breast cancer cells with Polyphenon-60 inhibited growth and decreased miR-27a expression [162]. As miR-27a has been shown to promote cancer cell proliferation in osteosarcoma cells [197], suppressing miR-27a may help these polyphenols to have anticancer effects. Downregulation of Sp1 may be linked to VEGF downregulation [22342309, 29048687], which can also explain the anticancer effects of these polyphenols.

Antitumor GEN has been shown to suppress miR-27a expression in pancreatic cancer cells [170]. Inhibiting miR-27a induced cell growth inhibition and apoptosis, implying that miR-27a is involved in GEN’s anticancer effect. Similarly, Xu et al. [171] discovered that GEN inhibited ovarian cancer cell growth and migration with downregulating miR-27a expression and increasing the expression of Sprouty2, a putative miR-27a target gene. GEN was also shown to inhibit uveal melanoma cell growth, which was accompanied by a decrease in miR-27a and an increase in its target gene ZBTB10 [172].

Apoptosis induction enhanced by miR-27a downregulation may be explained by its effect on caspase-9 activation through Apaf-1 upregulation, as demonstrated by experiments in which miR-27a antioligonucleotides promoted the formation of Apaf1-caspase-9 complex in TRAIL-treated colorectal cancer stem cells [198]. Yang et al. [173] reported that GEN have anticancer effects in lung cancer A549 cells by upregulating miR-27a and downregulating the proto-oncogene MET. The reason for the disparity in the results on GEN’s modulation of miR-27a is currently not known, but it could be due to the use of different cancer cells.

### 5.5. miR-93

CGA inhibited hepatoma and lung cancer cells by causing cell cycle arrest at the G_0_/G_1_ phase. Transfection of CGA-treated cells with mimics of miR-93 cancelled the p21 upregulation effect of CGA while increasing c-Myc, indicating that p21 is the target of miR-93 as reported in experiments for miR-20a [152].

EGCG inhibited cell growth and induced apoptosis in malignant neuroblastoma SK-N-BE2 and IMR-32 cells by decreasing Bcl-2 expression, increasing Bax expression, and activating caspase-8 and caspase-3 [129]. miR-92, 93, and 106b were downregulated by EGCG, while miR-7-1, miR-34a, and miR-99a were upregulated. miR-93 overexpression prevented EGCG-induced apoptosis, which was accompanied by an increase in Bcl-2 expression and a decrease in caspase-8 and caspase-3 activation. The findings suggest that miR-93 plays a role in EGCG-mediated apoptosis. Similarly, in neuroblastoma SH-SY5Y and SK-N-DZ cells, EGCG caused the downregulation of oncogenic miR-92, 93, and 106b and upregulation of tumor-suppressing miR-7-1, 34a, and 99a [130]. Prolonged exposure to estrogen is known to increase the risk of breast cancer [199]. Singh et al. [174] demonstrated that RES inhibited mammary carcinogenesis in a rat model of 17-estradiol-induced mammary tumors. Hormone-treatment induced increased tumor formation and expression of miR-93 in mammary tissues compared to control levels. The RES treatment had no effect on miR-93 expression levels.

### 5.6. miR-106b

CGA inhibited hepatoma and lung cancer cells by causing cell cycle arrest at the G_0_/G_1_ phase and transfection of CGA-treated cells with mimics of miR-106b reduced the CGA’s upregulation effect of p21 while increasing c-Myc [152].

As previously stated, EGCG inhibited the growth of malignant neuroblastoma cells, induced apoptosis, and reduced the expression of oncogenic miR-92, 93, and 106b [129,130]. In prostate cancer, RES exhibited anticancer activity and miR microarrays revealed that RES downregulated 23 miRs and upregulated 28 miRs [156]. Downregulation of miR-106b was confirmed by qRT-PCR. PTEN is one of the targets of downregulated miRs, including miR-106b and RES upregulated PTEN, suggesting that downregulation of miR-106b can lead to PTEN upregulation in the anticancer effect of RES. This notion is clearly demonstrated by Dhar et al. [154], who showed that RES decreased the levels of miR-17, miR-20a. and miR-106b, leading to upregulation of their target PTEN in prostate cancer cells. PTEN protein expression was downregulated when miR-106b was overexpressed, but it was upregulated in the presence of RES, indicating that PTEN is a direct target of miR-106b.

### 5.7. miR-155

CGA downregulated NK-κB and the nucleotide-binding domain like receptor protein 3 inflammasome-related proteins in a model of inflammation using LPS/ATP-stimulated RAW264.7 cells, which was dependent on the downregulation of miR-155 expression [21]. Ma et al. [30] showed that CUR suppressed LPS-induced cytokines (TNF-α, IL-6) and miR-155 expression in Raw264.7 and THP-1 cells in a similar experiment. Transfection of miR-155 mimics suppressed these effects, indicating that CUR suppresses LPS-induced inflammatory response by inhibiting miR-155. In experiments using a similar inflammation model, QUE was shown to downregulate cytokines such as TNF-α, IL-1β, and IL-6, as well as miR-155 [176]. Tili et al. [177] discovered that pretreatment with RES reduced the upregulation of miR-155 in LPS-treated THP-1 cells. As the results of several studies indicate a correlation between elevated levels of miR-155 and the development of tumors such as breast, lung, or gastric cancers, as well as leukemias, RES may be useful as an anti-inflammatory and anticancer agent. In metastatic breast cancer cells, GEN reduced cell viability and induced apoptosis by downregulating miR-155, FOXO3, PTEN, casein kinase, and p27. Overexpression of miR-155 in cells infected with miR-155 lentiviral vectors reduced the effects of GEN [175].

### 5.8. miR-221

Sarkar et al. [178] discovered that pancreatic cancer patients with high miR-221 expression have a lower rate of survival. Transfection of an miR-221 inhibitor suppressed pancreatic cancer cell growth while also upregulating PTEN, p27, and p57. A curcumin analogue CDF and isoflavone mixture containing 70.54% GEN mimicked the miR-221 inhibitor.

CUR reduced tumor weight and tumor microvessel count in a xenograft model inoculated with HepG2 cells compared to a vehicle control [24]. CUR decreased miR-221 expression while increasing miR-222 expression. miR-221 may be a target of anticancer strategies because it is involved in the angiogenesis mechanism. Expression of the tumor suppressor gene aplysia ras homolog I (ARHI) was found to be inversely associated with the expression of miR-221 and 222 in prostate cancer cell lines [180]. ARHI expression was significantly induced by transfection of miR-221 and 222 inhibitors. GEN upregulated ARHI expression in these cells by downregulating miR-221 and 222.

Wang et al. [181] discovered that QUE reduced LPS-induced inflammatory damage in WI-38 lung fibroblasts by increasing cell viability, suppressing cell apoptosis, and decreasing the production of inflammatory cytokines IL-6 and TNF-α. QUE inhibited LPS-induced upregulation of miR-221 in these cells, and miR-221 overexpression reversed QUE’s anti-inflammatory effects. Through downregulation of miR-221, QUE inhibited NF-B activity and the JNK pathway in LPS-treated cells. In human hepatocellular carcinoma HepG2 cells, EGCG inhibited cancer cell growth and induced apoptosis [115]. miR-let-7a, 16, and 221 were upregulated while miR-18a, 34b, 193b, 222, and 342 were downregulated, according to a microarray analysis and qRT-PCR results. Tumor-promoting effects of the minor upregulation of oncogenic miR-221 may be overcome by increased expression of tumor suppressive miR-16 and/or the anticancer effects of other miRs’ modulation, leading to EGCG’s eventual anticancer effects [115].

## 6. Conclusions

Consumption of coffee, tea, wine, curry, and soybeans has been linked to cancer prevention in epidemiological studies. A number of cell-based and animal studies have shown that polyphenols such as CGA, CUR, EGCG, GEN, QUE, and RES are major contributors to anticancer effects. Depending on their cellular microenvironments, these dietary polyphenols can act as both an antioxidant and a prooxidant, and several mechanisms have been proposed to explain their anticancer effects, one of which is an ROS-mediated mechanism (Figure 2). Furthermore, these polyphenols have been shown to modulate miRs expression. In general, they can increase the expression of tumor-suppressive miRs while decreasing the expression of oncogenic miRs, resulting in modulation of the expression/activity of constituents in ROS-mediated anticancer pathways (Figure 2) [3]. As a result, modulations by these miRs may enhance the anticancer effects of polyphenols in an additive or synergistic manner. In addition, other mechanisms such as EMT modulation by miRs may be involved in the anticancer effects of these polyphenols.

Several xenograft experiments such as those described above have shown that polyphenols modulate miRs in vivo [32,164,165]. However, only a few human studies have been conducted on this subject. miR-21 in the plasma of postmenopausal women with low bone density after CUR supplementation [200], miR-17, 27, and 146a in regulatory T cells from inflammatory rheumatic disease patients treated with CUR [201], and inflammation-responsive miRs such as miR-21, 34a, and 155 in peripheral blood mononuclear cells from type 2 diabetes and hypertensive patients who consumed RES-enriched grape extract [202] are just a few examples. Similar future studies in humans will provide convincing information on the effects of dietary polyphenols on cancer and other diseases.

A limitation of this review is that something other than what we have shown here may be found, as the results were obtained from a search of two databases: PubMed and Web of Science.

## Figures and Tables

**Figure 1 molecules-27-03816-f001:**
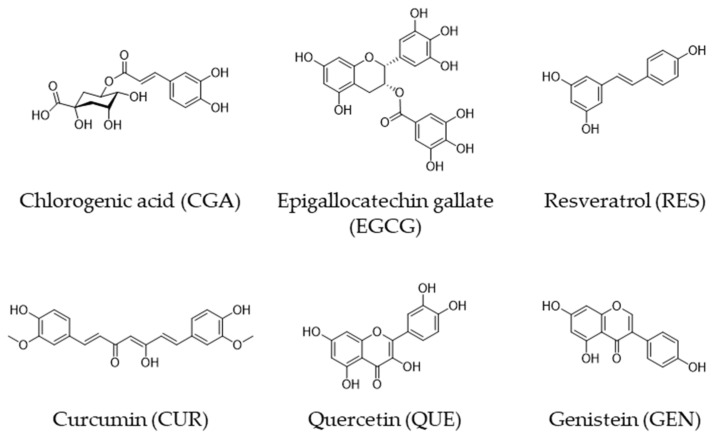
Chemical structures of CGA, EGCG, RES, CUR, QUE, and GEN.

**Figure 2 molecules-27-03816-f002:**
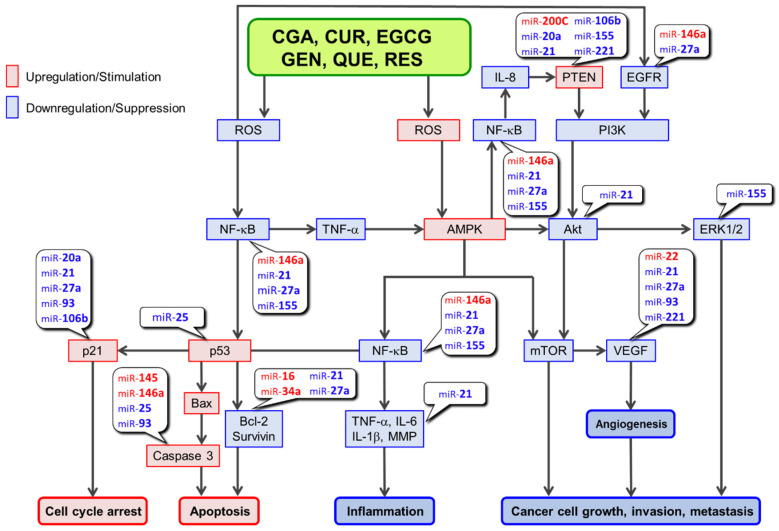
ROS-mediated anti-cancer activities associated with miRs regulated by polyphenols.

**Table 1 molecules-27-03816-t001:** Major food sources of polyphenols.

Polyphenol	Major Food Source
Chlorogenic acid (CGA)	Coffee bean
(−)-Epigallocatechin gallate (EGCG)	Green tea
Resveratrol (RES)	Red wine
Curcumin (CUR)	Curry
Quercetin (QUE)	Onion
Genistein (GEN)	Soy

**Table 2 molecules-27-03816-t002:** Modulatory effects of CGA, CUR, GEN, EGCG, QUE, and RES on ROS, AMPK, and NF-κB.

	ROS Up	AMPK Up	ROS Down	NF-κB Down
**Polyphenols**	Stimulation/ upregulation	Stimulation/ upregulation	Suppression/ downregulation	Suppression/ downregulation
**CGA**	Rakshit et al. [44] Hou et al. [55] Yang et al. [66]	Sudeep et al. [77] Lukitasari et al. [88] Santana-Galvez et al. [94]	Cha et al. [95] Wang et al. [96] Santana-Galvez et al. [94]	Zeng et al. [21] Chen et al. [34] Zatorski et al. [35]
**CUR**	Nakamae et al. [36] Gupta et al. [37] Gersey et al. [38]	Yu et al. [39] Hamidie et al. [40] Pan et al. [41]	Abadi et al. [42] Park et al. [43] Wang et al. [45]	Pimentel-Gutierrez et al. [46] Zhou et al. [47] Shao et al. [48]
**GEN**	Lee et al. [49] Zhang et al. [50] Park et al. [51]	Gasparrini et al. [52] Ikeda et al. [53] Lee et al. [54]	Cai et al. [56] Lee et al. [57] Lagunes et al. [58]	Mukund et al. [59] Mukund et al. [60] Javed et al. [61]
**EGCG**	Wei et al. [62] Ouyang et al. [63] Yang et al. [14]	Yang et al. [64] Ouyang et al. [63] Kim et al. [65]	Na et a. [67] Yang et al. [14] Wada et al. [68]	Shen et al. [69] Reddy et al. [70] Ohishi et al. [71]
**QUE**	Kim et al. [72] Lagunes et al. [58] Wang et al. [73]	Kim et al. [72] Zhang et al. [74] Fukaya et al. [75]	Bahar et al. [76] Priyadarsini et al. [78] Rezaei-Sadabady et al. [79]	Bahar, et al. [76] Cheng et al. [80] Chen et al. [81]
**RES**	Costa et al. [82] Fu et al. [83] Li et al. [84]	Wang et al. [45] Wang et al. [85] Baur et al. [86]	Giordo et al. [87] Perez-Torres et al. [89] Mathieu et al. [90]	Subedi et al. [91] Hsu et al. [92] Ginés et al. [93]

**Table 3 molecules-27-03816-t003:** Modulation by GEN and QUE of the molecules constituting the ROS-mediated anticancer pathway.

		GEN	QUE
p53	Upregulation	Ye et al. [97]	Priyadarsini et al. [101]
p21		Ye et al. [102]	Clemente-Soto et al. [103]
PTEN		Bilir et al. [104]	Boadi et al. [105]
EGFR	Downregulation	Gao et al. [106]	Pani et al. [107]
ERK		Li et al. [108]	Pan et al. [98]
VEGF		Yazdani et al. [99]	Lai et al. [100]
Bcl-2		Zhang et al. [50]	Pan et al. [98]

## Data Availability

Not applicable.

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
