# Peer review of "Anti-Cancer Effects of Dietary Polyphenols via ROS-Mediated Pathway with Their Modulation of MicroRNAs"

_molecules, 2022, doi:10.3390/molecules27123816_

Round 1
Reviewer 1 Report
1. The authors should mention which of the ROS biomarkers are getting modulated in Table 1. Just mentioning the names ( CGA, CUR, GEN, EGCG, QUE and RES on ROS, AMPK, and NF-κB) of the biomarkers in the title is not sufficient. Furthermore, please add the organism system (in vivo vs. in vitro), concentration and route of administration of each phytochemicals in both Table 1 and 2.
2. Authors are required to mention the concentrations of phytochemicals in both Table 3 and 4.
3. Please add either the author, year or only year in the tables. It is difficult to the tables with two modes references.
4. The authors should also the structure-function activity under the conclusions and speculate how enhancing the structures of the phytochemicals would effect their effects with different microRNA and the final outcomes in different cancer pathways.
5. Please cite Biomed Pharmacother. 2018 Nov;107:1648-1666. doi: 10.1016/j.biopha.2018.08.100. Epub 2018 Sep 8. PMID: 30257383.
Author Response
- The authors should mention which of the ROS biomarkers are getting modulated in Table 1. Just mentioning the names (CGA, CUR, GEN, EGCG, QUE and RES on ROS, AMPK, and NF-κB) of the biomarkers in the title is not sufficient. Furthermore, please add the organism system (in vivo vs. in vitro), concentration and route of administration of each phytochemicals in both Table 1 and 2.
Thank you for reviewer’s constructive suggestions. We revised both Tables which were Table 2 and 3, respectively in revised manuscript, since Table 1 was newly added. Most of ROS assay had been performed using H2DCF-DA, which reacts with not only H2O2 but also other peroxides. Therefore most studies had not identified ROS species. Additionally, some references in these Tables were Review articles, which contained no information about the concentrations. Therefore, it’s difficult to create an orderly Tables. Some information about concentrations were described in the main text.
- Authors are required to mention the concentrations of phytochemicals in both Table 3 and 4.
These tables have been handled in the same manner as above.
- Please add either the author, year or only year in the tables. It is difficult to the tables with two modes references.
We showed only author names; therefore, year was deleted from tables.
- The authors should also the structure-function activity under the conclusions and speculate how enhancing the structures of the phytochemicals would effect their effects with different microRNA and the final outcomes in different cancer pathways.
We have already discussed structures and mechanisms of polyphenols-induced miRNA regulations associated with anti-cancer activities (in Conclusion section).
- Please cite Biomed Pharmacother. 2018 Nov;107:1648-1666. doi: 10.1016/j.biopha.2018.08.100. Epub 2018 Sep 8. PMID: 30257383.
We cited above article.
Reviewer 2 Report
I commend the authors: Yasukiyo Yoshioka , Tomokazu Ohishi, Yoriyuki Nakamura , Ryuuta Fukutomi and Noriyuki Miyoshi of the manuscript titled “Anti-Cancer Effects of Dietary Polyphenols via ROS-mediated Pathway with Their Modulation of MicroRNAs” for their review on the ROS-mediated anticancer properties of CGA, CUR, EGCG, and RES
Before this review is published, there are several things need to be addressed or corrected:
1- In the introduction, please insert a table stating major food sources (types of foods, including fruits, vegetabls, crops, medicinal plants, spices..their names, drinks such as tea types and coffee …) of each of investigated polyphenols
2- I feel the review is very long and need to be shortened.
3- In the conclusion, add the limitation of this review as well.
Author Response
I commend the authors: Yasukiyo Yoshioka , Tomokazu Ohishi, Yoriyuki Nakamura , Ryuuta Fukutomi and Noriyuki Miyoshi of the manuscript titled “Anti-Cancer Effects of Dietary Polyphenols via ROS-mediated Pathway with Their Modulation of MicroRNAs” for their review on the ROS-mediated anticancer properties of CGA, CUR, EGCG, and RES
Before this review is published, there are several things need to be addressed or corrected:
1- In the introduction, please insert a table stating major food sources (types of foods, including fruits, vegetabls, crops, medicinal plants, spices..their names, drinks such as tea types and coffee …) of each of investigated polyphenols
As suggested by reviewer, a new table was inserted as Table 1.
2- I feel the review is very long and need to be shortened.
As indicated by reviewer 1, some parts modification and shortening were attempted.
3- In the conclusion, add the limitation of this review as well.
A limitation of this review has been added at the end of the “Conclusion” section.
Round 2
Reviewer 1 Report
The authors have addressed the comments satisfactorily. The article is suitable for publication in "Molecules".
Reviewer 2 Report
Accepted for me